# Integrated System Pharmacology Approaches to Elucidate Multi-Target Mechanism of *Solanum surattense* against Hepatocellular Carcinoma

**DOI:** 10.3390/molecules27196220

**Published:** 2022-09-21

**Authors:** Hafiz Rameez Khalid, Muhammad Aamir, Sana Tabassum, Youssef Saeed Alghamdi, Ahmad Alzamami, Usman Ali Ashfaq

**Affiliations:** 1Department of Bioinformatics and Biotechnology, Government College University Faisalabad, Faisalabad 38000, Pakistan; 2Department of Biochemistry, Government College University Faisalabad, Faisalabad 38000, Pakistan; 3Department of Biology, Turabah University College, Taif University, Taif 21944, Saudi Arabia; 4Clinical Laboratory Science Department, College of Applied Medical Science, Shaqra University, Al Quwayiyah 11961, Saudi Arabia

**Keywords:** network pharmacology, *S. surattense*, molecular mechanism, hepatocellular carcinoma (HCC)

## Abstract

Hepatocellular carcinoma (HCC) is one of the most common malignant liver tumors with high mortality. Chronic hepatitis B and C viruses, aflatoxins, and alcohol are among the most common causes of hepatocellular carcinoma. The limited reported data and multiple spectra of pathophysiological mechanisms of HCC make it a challenging task and a serious economic burden in health care management. *Solanum surattense* (*S. surattense*) is the herbal plant used in many regions of Asia to treat many disorders including various types of cancer. Previous in vitro studies revealed the medicinal importance of *S. surattense* against hepatocellular carcinoma. However, the exact molecular mechanism of *S. surattense* against HCC still remains unclear. In vitro and in silico experiments were performed to find the molecular mechanism of *S. surattense* against HCC. In this study, the network pharmacology approach was used, through which multi-targeted mechanisms of *S. surattense* were explored against HCC. Active ingredients and potential targets of *S. surattense* found in HCC were figured out. Furthermore, the molecular docking technique was employed for the validation of the successful activity of bioactive constituents against potential genes of HCC. The present study investigated the active “constituent–target–pathway” networks and determined the tumor necrosis factor (TNF), epidermal growth factor receptor (EGFR), mammalian target of rapamycin (mTOR), Bcl-2-like protein 1(BCL2L1), estrogen receptor (ER), GTPase HRas, hypoxia-inducible factor 1-alpha (HIF1-α), Harvey Rat sarcoma virus, also known as transforming protein p21 (HRAS), and AKT Serine/Threonine Kinase 1 (AKT1), and found that the genes were influenced by active ingredients of *S. surattense*. In vitro analysis was also performed to check the anti-cancerous activity of *S. surattense* on human liver cells. The result showed that *S. surattense* appeared to act on HCC via modulating different molecular functions, many biological processes, and potential targets implicated in 11 different pathways. Furthermore, molecular docking was employed to validate the successful activity of the active compounds against potential targets. The results showed that quercetin was successfully docked to inhibit the potential targets of HCC. This study indicates that active constituents of *S. surattense* and their therapeutic targets are responsible for their pharmacological activities and possible molecular mechanisms for treating HCC. Lastly, it is concluded that active compounds of *S. surattense* act on potential genes along with their influencing pathways to give a network analysis in system pharmacology, which has a vital role in the development and utilization of drugs. The current study lays a framework for further experimental research and widens the clinical usage of *S. surattense*.

## 1. Introduction of Hepatocellular Carcinoma

HCC, a complex malignant tumor, is lethal and ranks sixth in mortality worldwide. The prevalence of HCC is still increasing broadly and poses a significant threat to humans [1]. This intricate condition of lethal disease is followed by multiple factors, including persistent viral infections, toxin carcinogenesis, cirrhosis due to fatty liver disease, and numerous genetic factors. HCC is linked with cirrhosis and typically results in the development of primary liver illnesses such as hepatitis B virus (HBV) and hepatitis C virus (HCV) infections [2,3]. The available therapeutic interventions against HCC are surgical excision, local ablation, trans-arterial chemoembolization (TACE), liver transplantation, and medical therapy with sorafenib [4]. In addition, other treatment paradigms include novel medications such as Lenvatinib, Regorafenib, and Ramucirumab, all proving significant ameliorative effects in advanced HCC patients. Despite these potent therapeutic interventions, the survival rate of HCC patients is still compromised [2,5]. For this reason, more effective alternative medicines with minimal toxicity should be developed to augment the overall survival of HCC patients. In this situation, exploring the preventive effects of herbal medicines against HCC seems a favorable option.

Herbal medicine has its origin in ancient traditions and cultures. Herbal medicine involves plants’ involvement to enhance general health and cure disease. These herbal medicines are a rich source of potent ingredients and can be taken as pharmaceutical medications. Even now, many pharmaceutical drugs can be derived from naturally occurring compounds found in plants. Traditional Chinese medicines (TCM) have been practiced in China for thousands of years and are the most prominent complementary and alternative therapy [6,7]. TCM positively modifies the state of human health from the diseased condition [8]. Many clinical studies on TCM have been shown to have an anti-cancer impact by causing cancer cell death, boosting the immune system, initiating cell differentiation, and preventing tumor development and metastasis [9]. According to the WHO, traditional medicine is used by 80% of the world’s population.

Consequently, there is increased attention to the usage of herbal medications around the globe [10]. *S. surattense*, family Solanaceae, genus Solanum, is a perennial herb, and it is widely considered the most valuable traditional remedy [11]. After extensive research on phytochemical ingredients, the alkaloid and steroidal compounds solanine, solamargine, campesterol, and diosgenin were isolated. Plant-derived secondary metabolites possess a wide range of biological properties beneficial for humans, such as anti-cancer, anti-inflammatory, antioxidant, and antiallergic activities. Due to the presence of phytochemicals such as apigenin, lupeol, solamargine, and diosgenin, the medicinal plant *S. surattense* possesses anti-cancer properties [12]. Bioactive chemicals derived from plants have always been important in developing therapeutic medications. However, the mechanism of the anti-cancer effect of *S. surattense* against HCC is still unknown.

Network pharmacology, the next paradigm in drug discovery, is a critical technique for evaluating in silico therapeutic interventions [13]. This integrated technique helps to identify the molecular mechanism, synergistic actions, and therapeutic target between traditional medicines and diseases. Network pharmacology analysis is a holistic strategy that focuses on different compounds/targets/one disease. Because of its multiple-dimension paradigm, approaches can narrate the underlying complexities among biological systems, diseases, and drugs from a network perspective. This powerful method has shifted the paradigm from a “one-disease/one-drug/one-target” mode to a “network-target, multiple-component-therapeutics” mode [14,15]. With this, a single ligand can bind multi-targets, and the benefits of the therapeutic medicinal herb would be the cheapest, most selective, and relatively non-toxic multi-target drug. Network pharmacology has effectively been used to screen active components and disclose the pharmacodynamic processes of CHM as an emerging subject in modern CHM pharmacological research.

Interestingly, network pharmacology has been recognized as a potent and promising approach for developing herbal medicine against HCC. Compared to the western medicine approach based on a one-disease/one-drug methodology, drug discovery still faces various issues including efficacy, safety, and sustainability [16]. This study applies bioinformatics data mining and a network pharmacology approach to find active constituents of *S. surattense* and their therapeutic targets responsible for their pharmacological activities and potential molecular mechanisms for the treatment of HCC.

## 2. Results

### 2.1. In Vitro Toxicological Analysis Hepatocellular Carcinoma and Normal Cells

The toxicological activity of *S. surattense* fruit extract at different concentrations (30 to 100 µg/mL) was explored in both liver hepatoma cells (HepG2 cells) and Vero cell lines (African green monkey’s kidney cells) in a dose-dependent manner. Among all of these concentrations of *S. surattense* fruit extract, including 60, 70, 80, 90, and 100 µg/mL, the results showed a less than 50% cell viability as compared to positive control doxorubicin. *S. surattense* fruit extract exhibited a non-toxic effect in Vero cells (Figure 1A,B). Our results showed that *S. surattense* fruit extract (from 60 to 100 µg/mL concentration) is highly effective against liver cancer as analyzed in in vitro analysis with a resultant IC_50_ value of 62.70 ± 1.14 µg/mL.

### 2.2. Active Compounds and Target Screening

Out of 73 extracted active ingredients of *S. surattense*, 20 ingredients were selected as they fulfilled the drug-likeness properties, molecular weight (MW < 500), logarithm of the partition coefficient (logP < 5.6), number of hydrogen bond acceptors (nHA ≤ 10), number of hydrogen bond donors (nHD ≤ 5), and the Lipinski rule of five. The three other compounds (solanine, Quercetin 3-Galactoside 7-Rhamnoside, and Quercetin 3-Sophorotrioside 7-Rhamnoside) were included in this study because their anti-cancer activity was checked in the lab. However, they did not meet the inclusion criteria (Table 1). A total of 756 targets of active ingredients were screened through the SwissTarget Prediction database. After searching, filtering, and removing the duplicates, 487 potential targets of hepatocellular carcinoma were screened from four databases (OMIM, DisGeNET, CTD, and GeneCard). Furthermore, 79 anti-cancer key targets of *S. surattense* were selected for further analysis.

### 2.3. Protein–Protein Network and Hub Genes

The target genes were imported into the STRING database for the PPI network. The relationship between many targets throughout disease development is depicted in the PPI network by nodes and their related interactions. Later, the Cytohubba plugin was used to find the hub genes in the network. Tumor protein 53 (TP53)-, (AKT1), ESR1, EGFR-43, TNF, HIF1A-, HRAS-, MTOR-41 and apoptosis regulator Bcl-X BCL2L1- showed the highest degree (Figure 2). The highest degree indicates that the targeted genes are highly connected, implying that all of them could be important targets. After correlating the outcomes to those received by KEGG analysis, eight targets AKT1, ESR1, EGFR, TNF, HIF1A, HRAS, MTOR, and BCL2L1, were chosen for further molecular docking analysis.

### 2.4. GO Enrichment Analysis

Potential biological functions of *S. surattense* targets were discovered using GO annotation and pathway enrichment studies. According to GO enrichment analysis, the key targets were associated with oxidoreductase activity, peptidyl–serine modification, nuclear receptor activity, etc. Using KEGG pathway analysis, relevant signaling pathways related to *S. surattense* anti-cancer action were identified. GO enrichment and KEGG analysis reveal that hub genes were significantly enriched genes (Figure 3).

### 2.5. Pathway Network Construction

The network between hub genes, active constituents, and KEGG-enriched pathways was constructed by Cytoscape (version 9). A network analyzer was used to analyze the network; 34 nodes and 74 edges were found in the compound target pathway network (Figure 4). The 74 edges show the interactions between anti-cancer targets of *S. surattense* and their pathways.

### 2.6. Molecular Docking

For molecular docking, from the Protein Data Bank (PDB) database, MTOR (1AUE), EGFR (1IVO), BCL2L1 (1R2E), ESR1 (1UOM), HRAS (4XVR), HIF1A (5JWP), TNF (5MU8), and AKT1 were considered as the three-dimensional structures of the eight top targets to be selected. Auto Dock (V. 4.2) was employed to validate the best possible binding modes of the interactions. The ligand was supposed to be flexible during docking, whereas the protein was rigid. All eight active components from *S. surattense* were docked with the five potential targets of HCC. The 2D structures of these eight active compounds are shown in Figure 5. The docking score and binding energy were critical criteria for screening components (Table 2). Clusters with the most remarkable confirmation and absolute binding energy value were selected. The docking analysis was validated by other software: MOE and Pyrx. Docked complexes of potential targets with active constituents almost showed the same binding affinity and RMSD value by different sources Figure 6. In the molecular docking, the active ingredients showed a high binding energy and Rmsd with the potential targets, which suggested that active ingredients of *S. surattense* have better binding interaction with HCC targets. Figure 5 shows the molecular docking complex of targets and the high binding score.

## 3. Discussion

HCC is among the most common health disorders that significantly impact the world population. According to GLOBOCAN, the highest occurrence rate of HCC has been reported in China, Southeast Asia, Eastern Africa, and Sub-Saharan Western Africa. The interlinked series of multiple pathological pathways comprehending the HCC further intensify the disease and make treating it more challenging [17]. Despite the continuous efforts of medical health care practitioners in drug development to subdue HCC, it still lacks any effective therapeutic intervention. Thus, exploring any effective therapeutic compound or drug is crucial. Continuing with the same notion, traditional medicinal plants have been inclusively practiced for their effective contribution to many diseases, including HCC, based on their multiple pharmacological properties. Continuing with the same notion, traditional medicinal plants have been inclusively practiced for their effective contribution to many diseases, including HCC, based on their multiple pharmacological properties. For example, *Calotropis gigantea* (Asclepiadaceae) and *Xanthium strumarium* (Asteraceae) have anti-hepatocellular carcinoma and anti-proliferative (against HepG2 cancer cells) properties, respectively [18].

The medicinal plant may have broad pharmacological actions with multiple targets and pathways composed of multiple components that play a vital role in treating HCC [19]. Natural products and their derivatives account for approximately half of all clinically used medicines. They have been a prominent topic in recent years as a research trend and potential source for targeted therapies due to their structural variety, multi-target action, and minimal toxicity side effects [20]. High throughput approaches have shown significant modes in screening the pharmacological effectiveness of herbal remedies in drug development in the last decade [21,22].

The network pharmacology (NP) approach, which associates biological systems with in silico technology, might predict potential targets and pathways underlying the anti-cancer effect of HCC. In this current study, we employed NP to reveal the pharmacological mechanism of active constituents in *S. surattense* against HCC. Compounds with a high degree in the compound–target network may consider most *S. surattense* therapeutic effects on HCC.

The NP approach could help us to screen the putative active ingredients and target ingredients. *S. surattense,* the identified compound with OB 30% and DL index v0.18, was regarded as a potentially bioactive compound because it is probably absorbed and distributed in the human body. In this context, the screening results revealed that glycoalkaloid, steroids, triterpenoids, solanine, and flavonoids were the main phytochemical bioactive compounds of *S. surattense* that proved potent in the development of HCC by affecting the TP53, GAPDH, AKT1, ESR1, EGFR, TNF, HIF1A, HRAS MTOR, and BCL2L1 gene targets. The PPI network analysis was performed to identify essential proteins associated with the anti-HCC effects of *S. surattense*, which were indicated by the PPI network’s critical nodes. The interactive nodes showed that *S. surattense* might interact with representative gene targets directly or indirectly to perform anti-cancer actions. These key targets were mainly involved in biological processes such as tumor cell proliferation, apoptosis, cell cycle control, epithelial–stromal transition, angiogenesis, tumor invasion and metastasis, tumor signal transmission, immunological modulation, drug resistance, and others [23,24,25]. Our present study’s predicted outcomes were consistent with certain earlier publications.

This study said that quercetin, followed by solanine, was the vital key ingredient of *S. surattense.* Quercetin, a natural flavonoid, was the most potent compound that could suppress the growth of HCC cells employed in aerobic glycolysis, and suppress the development of HCC by lowering HK2 protein levels and inhibiting the AKT/mTOR pathway in HCC cells [26]. The solanine, a class of glycoalkaloids, induces apoptosis by regulating the expression of TNF, decreasing the expression of anti-apoptotic proteins (Bcl-2), increasing the expression of apoptotic proteins (Bax), reducing the Bcl-2/Bax ratio, obstructing mitochondrial and lysosomal membrane proteins, and activating the caspases cascade. Furthermore, solanine decreases the phosphorylation of ERK1/2, JNK1/2, and PI3K/Akt signaling pathways [12,27,28].

As shown in GO term and KEGG analysis using enrichment pathways, these findings, mainly linked with the GO results, suggested that *S. surattense* may reduce HCC cell growth and induce cell death via acting on the PI3K-AKT signaling pathway, which represents the putative mechanism of action of HCC growth. Further studies confirmed that quercetin significantly reduced the expression of p-PI3K p85 (Tyr467), p-PI3K p55 (Tyr199), and p-Akt (Ser473) in HepG2. Quercetin, as a potent flavonoid component in *S. surattense* associated with an effective anti-cancer drug, may boost the expression of p53 which in turn downregulates the expression of HIF1A, resulting in the proapoptotic process of liver cancer [29].

Additionally, activating the PI3K-AKT pathway can modulate HIF1A expression [30]. According to a series of network pharmacology studies, *S. surattense* may have inhibition in HCC via directly influencing cancer, metabolic, and immune-related pathways [31]. EGFR, a transmembrane receptor tyrosine kinase, might activate various signaling pathways that control cell proliferation, differentiation, and survival [32]. AKT1 is related to tumor development and metastasis in HCC and is activated by the PI3K pathway. AKT2 overexpression is prevalent in HCC and is associated with a bad prognosis [33,34]. The results indicated that *S. surattense* could inhibit the expression of AKT1 and phosphorylation of AKT1. It was predicted that *S. surattense* controls several targets and pathways in HCC cells based on our findings (Figure 7). Molecular docking was employed to validate further, based on the spatial structure of ligands and receptors for estimating binding energy between medicinal constituents and targets [35]. In this study, we discovered several target genes that are involved in various metabolic pathways.

In the context of network pharmacology, the current study elucidates the active compounds, their potential targets, and the related pathways against HCC. Consequently, it provides a theoretical framework for subsequent experimental investigation. Network pharmacology has its limitations; therefore, basic pharmacological pathways for treating HCC can only be determined by data mining. Although we have given some intriguing facts, further research and clinical trials are necessary to verify the therapeutic uses of *S. surattense.*

## 4. Material and Methods

### 4.1. Collection and Identification of S. surattense

The *S. surattense* was collected from the Cholistan Desert, Bahawalpur, Pakistan, and its identification was confirmed by the Director Botanical Science Division, Pakistan Museum of Natural History Islamabad, Pakistan (Herbarium Number 043720).

### 4.2. Extraction of Plant Material

The *S. surattense* extract preparation was performed following the maceration method [35]. In brief, the fruit part of *S. surattense* was shade-dried at room temperature and ground with an electric grinder to obtain a coarse powder. Then, 50 g of *S. surattense* powder was soaked in 500 mL of 70% ethanol (1:10 *w*/*v*; plant material/solvent), vigorously agitated 2X/day for seven consecutive days, and filtered using Whitman’s filter paper. The filtrate was dried in a hot air oven at 40 °C to obtain a solid plant extract and stored at −4 °C in amber glass vials until use. There are no reports of such studies on *S. surattense* to date. Therefore, we wanted to obtain a non-toxic extract that might contain a maximum number of phytochemicals of different classes, and which would not lose its biological activities due to beyond-physiological temperatures. For this purpose, maceration (cold extraction method) was used, although the quantity of phytochemicals might have been compromised by said method, but not the quality. Likewise, 70% ethanol (relatively highly polar and nontoxic solvent) was used instead of methanol (relatively toxic solvent) or a nonpolar solvent to obtain an extract from plant material, so that number of biologically active phytochemicals with their relatively nontoxic properties could be extracted [36,37]. The final yield of obtained extract (10.5 mg) was 21%.

### 4.3. Preparation of Stock Solutions

A stock plant extract solution was prepared in dimethyl sulfoxide (DMSO) with a 50 mg/mL concentration, filtered using a 0.22 μm syringe filter, and stored at −20 °C until further experimentation [38].

### 4.4. In Vitro Assay and Cell Culture Medium

Human liver cells HepG2 and Vero (ATCC, Manassas, VA, USA) were maintained in a high-glucose DMEM medium containing 10% fetal bovine serum, 1% penicillin, and 1% streptomycin, then cultured at 37 °C in an atmosphere with 5% CO_2_ in a carbon dioxide incubator [39,40]. To subculture, the spent media were removed from the flask, cells were washed by DPBS, and then they were incubated with trypsin EDTA (1 mL/25 cm^2^) at 37 °C for 2–5 min. Trypsin was inactivated after detaching the cells by adding an FBS-containing medium [41], and counted by a hemocytometer to determine the concentration of cell suspension.

### 4.5. Cell Proliferation Assay/Toxicological Analysis of Plant Extract Using WST-8 Assay

The HepG2 and Vero cell lines were grown in a 96-well cell culture plate up to 70% confluence in respective media for 24 h. Subsequently, the cells were treated with different concentrations (30, 40, 50, 60, 70, 80, 90, and 100 µg/mL) of *S. surattense* fruit extract and vehicle (DMSO control) for another 24 h. The cells were washed using a phosphate buffer saline (PBS) following the media aspiration. The WST-8 solution was added (10 μL/well in 100 μL DMEM) [42,43], and the plate was incubated for 4 h at 37 °C in the dark. The water-soluble, orange-colored WST-8 formazan product was analyzed by reading the absorbance using an ELISA plate reader at a test wavelength of 450 nm [44] and a reference wavelength of 630 nm. Each experiment was repeated for at least 4X prior to the data plotting a graph. Meanwhile, the untreated cells were considered the control [45], and Doxorubicin (1.2 µg/mL)/(2.21 µM) was used as a positive control [46].

Cell viability/proliferation% was evaluated by the following equation:$$\mathrm{Cell}\;\mathrm{Proliferation}\%=\frac{\mathrm{Abs}.\;\mathrm{of}\;\mathrm{Sample}-\mathrm{Abs}.\;\mathrm{of}\;\mathrm{Reference}\;\left(\mathrm{Sample}\right)}{\;\mathrm{Abs}.\;\mathrm{of}\;\mathrm{Control}-\mathrm{Abs}.\;\mathrm{of}\;\mathrm{Reference}\;\left(\mathrm{Control}\right)}\times 100$$

#### Screening of Active Compounds and Targets

The active components of *S. surattense* were identified using PubChem, Swiss ADME, and canonical SMILES of bioactive compounds were collected from ChEMBL. Using the obtained canonical SMILES, five pharmacokinetic properties of bioactive compounds, including molecular weight, hydrogen bond acceptor, hydrogen bond donor, and the logarithm of partition coefficient, were obtained. The Lipinksi rule of five for drug-likeness was also analyzed to screen for these bioactive compounds. Following the acute accuracy, a comprehensive ADMET analysis was performed using ADMETlab. Bioactive compounds were screened out based on the Lipinksi rule of drug discovery, including the criteria of drug-likeness, molecular weight (MW < 500), the logarithm of the partition coefficient (logP < 5.6), number of hydrogen bond acceptor (nHA ≤ 10), and number of hydrogen bond donor (nHD ≤ 5). Further, SwissTarget Prediction was used to select the potential target of the selected compounds using canonical SMILES. To predict the target of selected compounds, the *Homo sapiens* species was determined.

### 4.6. Prediction of HCC Targets

Predicting disease-related genes is a first step toward understanding the molecular mechanisms of medicinal herbs for treating various diseases and syndromes. The four disease-related databases, GeneCard, OMIM, CTD, and DisGeNET, were searched and evaluated to predict possible disease-related targets using hepatocellular carcinoma as a keyword. Furthermore, each database contains brief genomic information and functional annotations for all known human genes. All duplicated genes were removed for further analysis.

### 4.7. Screening of Key Targets

The identification of overlapping disease and chemical targets was made using Venn diagrams. Those are deemed important targets and are subjected to further investigation. Protein–protein interactions (PPI) are significant because of their flexibility, suppleness, and selectivity. STRING is a widely used database of protein–protein interactions that includes information from various sources. The common key targets of active compounds and HCC disease were uploaded to the STRING database for interaction analysis of the key targets at a combined score of 0.4. To visualize the protein–protein interaction network, Cytoscape (version 9) was used [47].

### 4.8. Analysis of Functional Enrichment

KEGG pathway analysis and GO enrichment analysis was performed using DAVID. The common key targets of disease and plant were subjected to the DAVID database for the functional enrichment analysis. Three GO annotations, cellular components (CC), biological process (BP), and molecular function (MF) were analyzed. KEGG was used for the pathway analysis of the key targets. The top 10 GO annotations and cancer-related KEGG pathways were selected with P-value less than 0.05, and these were discussed in the article in the form of dot plots made by the R ggplot2 package.

### 4.9. Network Construction

The molecular mechanism of *S. surattense* in HCC was studied through network analysis. The active components of *S. surattense* and the HCC therapeutic target were entered into Cytoscape to form the compound–target network. The network nodes represent the chemical ingredients and targets, while the edges represent their interactions. The “network analyzer” examined the network’s essential parameters.

### 4.10. Molecular Docking

In drug discovery, molecular docking has become a significant catalyst and the most applicable technique. Through molecular docking, it is possible to predict the interaction of ligands to their respective proteins in a crystalline lattice. The crystal structures of proteins MTOR (1AUE), EGFR (1IVO), BCL2L1 (1R2E), ESR1 (1UOM), HRAS (4XVR), HIF1A (5JWP), TNF (5MU8), and AKT1 were retrieved from the RCSB PDB database. The Protein Data Bank (PDB) is a globally interconnected repository for information on the three-dimensional structures of proteins and nucleic acids. Researchers may use these data to better understand various facets of biomedical, from the synthesis of protein to disease. Chimera was used to refine the structure of the protein. Furthermore, molecular docking between key targets and active molecules was done using PyRx software. The docking score between key targets and compounds was used as a primary evaluation factor to filter out potential constituents and their potential targets. The docking analysis was also performed with MOE and Autodock software to validate the docked structures. Then Chimera and Discovery Studio were used to visualize interactions between key targets and active compounds. By measuring the strong affinity between chemicals and their related targets, this stage attempted to investigate the binding energy among compounds.

## 5. Conclusions

According to a retrospective study, we conclude that herbal medicine (HM) therapy was correlated with a better prognosis in patients with HCC. Furthermore, by predicting active components and molecular targets of herbs, the network pharmacology approach may be used to explore the underlying anti-cancer mechanisms of herbs. A potential research strategy for “precise HM treatment” is the integration of clinical studies with network pharmacology. This study also concludes that the active constituents of *S. surattense* such as quercetin control the disorder by suppressing the potential target of hepatocellular carcinoma. The anti-cancer effects of *S. surattense* on HCC were postulated to be associated with the regulation of tumor cell proliferation, apoptosis, angiogenesis, tumor invasion, and metastasis via multiple signaling pathways, including the AKT1, EGFR signaling pathway, the ESR1 signaling pathway, and the HIF1A signaling pathway.

## Figures and Tables

**Figure 1 molecules-27-06220-f001:**
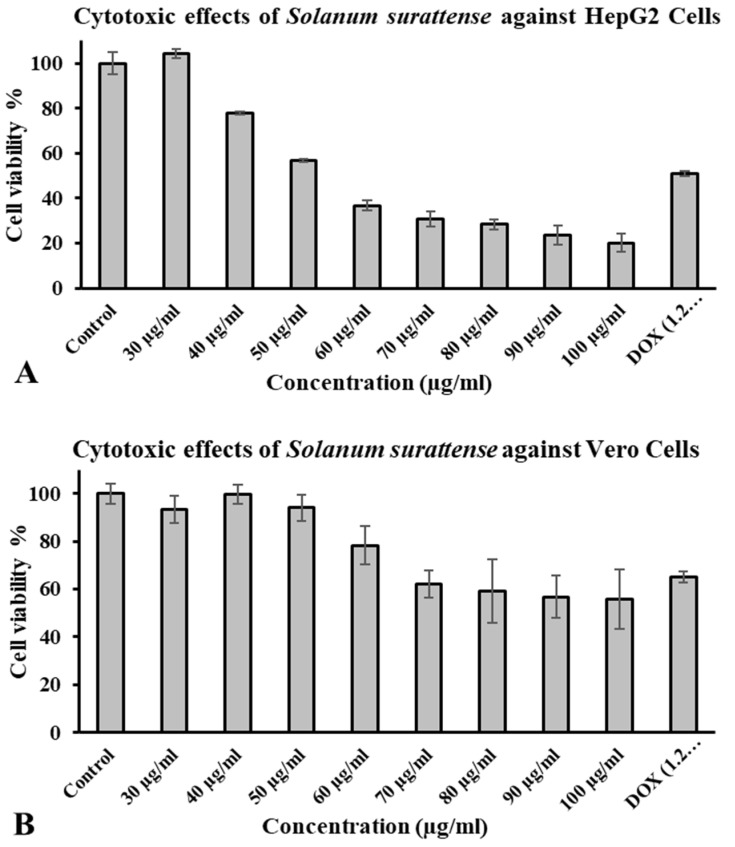
Cell viability % of *S. surattense* fruit extract and Doxorubicin as a positive control. (**A**) Anti-cancer activity of *S. surattense* fruit extract on HepG2 cell viability. HepG2 cells treated with different concentrations (30, 40, 50, 60, 70, 80, 90, and 100 µg/mL) of *S. surattense* fruit extract (*x*-axis) and cell viability % relative to control (*y*-axis). The extract concentrations (40, 50, 60, 70, 80, 90, and 100 µg/mL) were significant with statistical difference (*p* < 0.00) and STDs 2.1, 3.2, 2.2, 4.2, and 3.85 for respective extracts. (**B**) Anti-cancer activity of *S. surattense* fruit extract on Vero cells viability. Vero cells treated with different concentrations (30, 40, 50, 60, 70, 80, 90, and 100 µg/mL) of *S. surattense* fruit extract (*x*-axis) and cell viability % relative to control (*y*-axis). The extract concentrations (60, 70, 80, 90, and 100 µg/mL) were significant with statistical difference (*p* < 0.00) and STDs 8.06, 5.7, 13.2, 8.74, and 12.4 for respective extracts.

**Figure 2 molecules-27-06220-f002:**
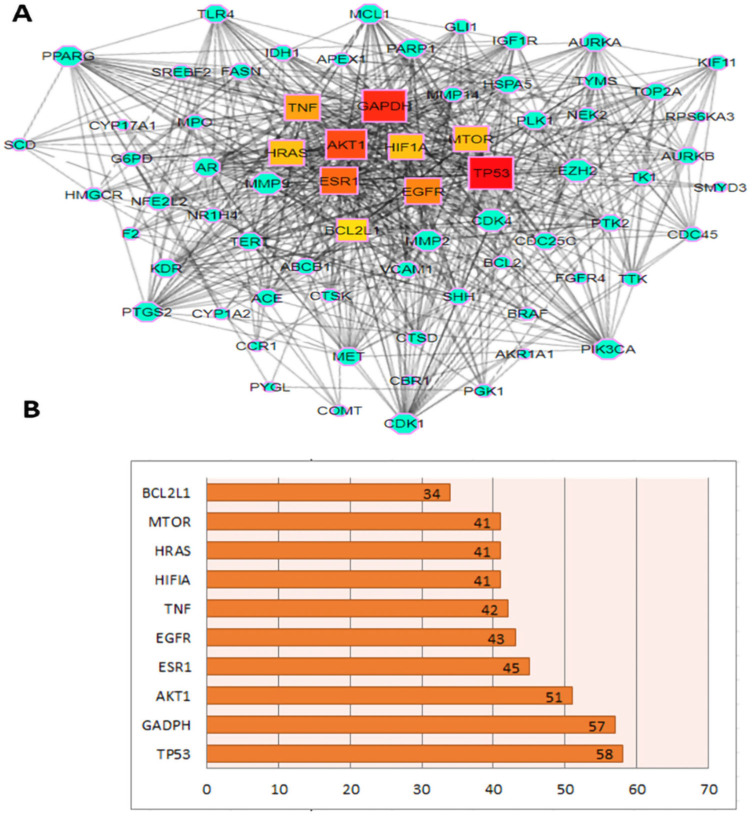
(**A**) PPI network analysis. Central square nodes represent the targets with a higher degree (hub genes), the color from red to yellow and node size set according to their degree score; the rest of the blue nodes were other key targets. (**B**) The bar plot drawn based on the degree score represents the hub genes.

**Figure 3 molecules-27-06220-f003:**
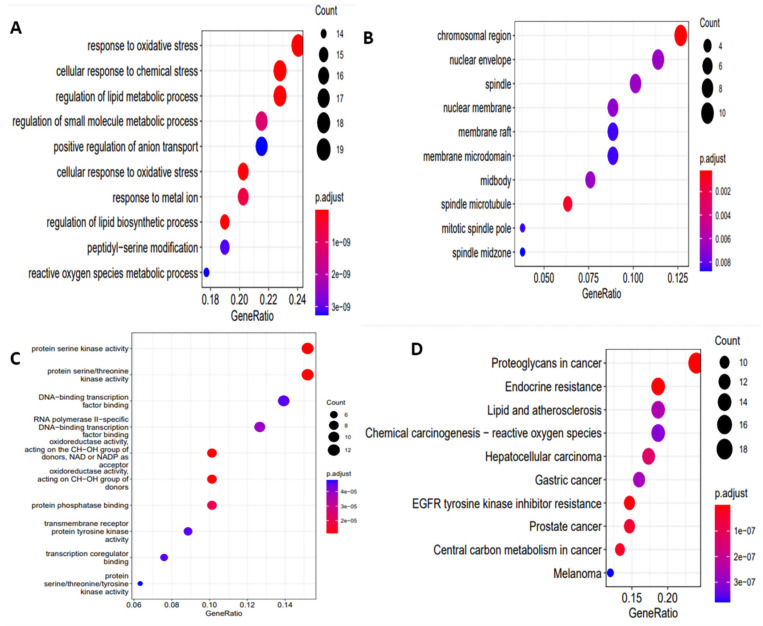
GO and KEGG enrichment analysis of key targets. (**A**) Biological process, (**B**) cellular components, (**C**) molecular functions, and (**D**) enriched KEGG pathways of key targets.

**Figure 4 molecules-27-06220-f004:**
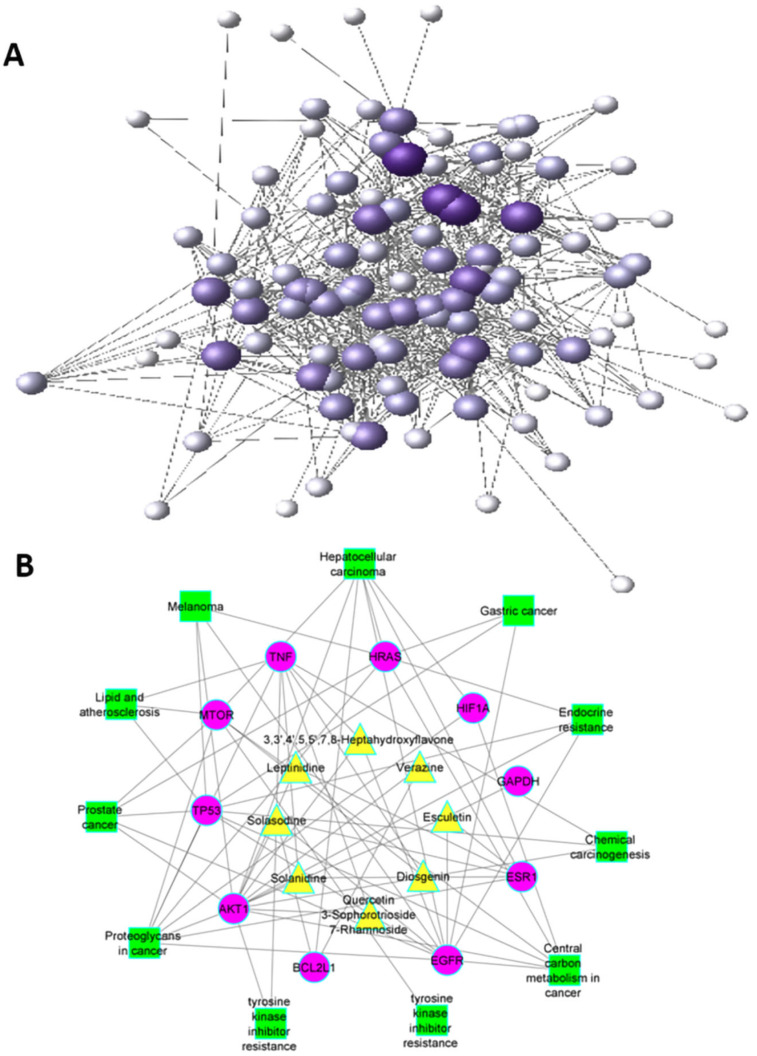
(**A**) Compound–target network. (**B**) Compound–target–pathway network; central yellow color triangles represent active compounds of *S. surattense,* hub gene represented by purple-colored circles, and outer green color squares represent modulating pathways.

**Figure 5 molecules-27-06220-f005:**
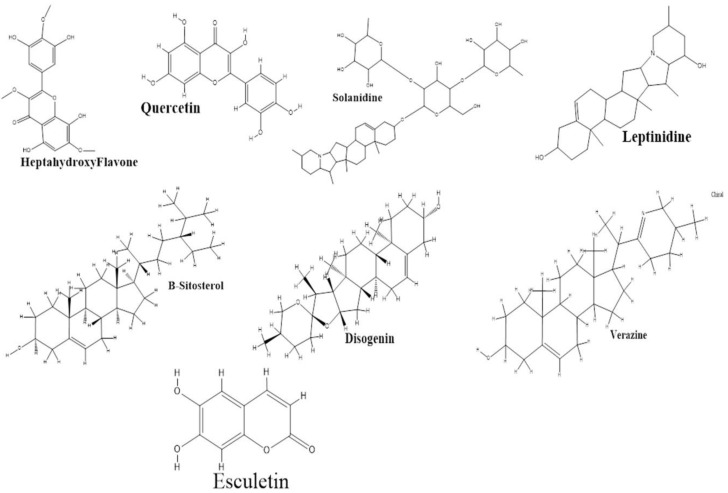
2D structures of active compound.

**Figure 6 molecules-27-06220-f006:**
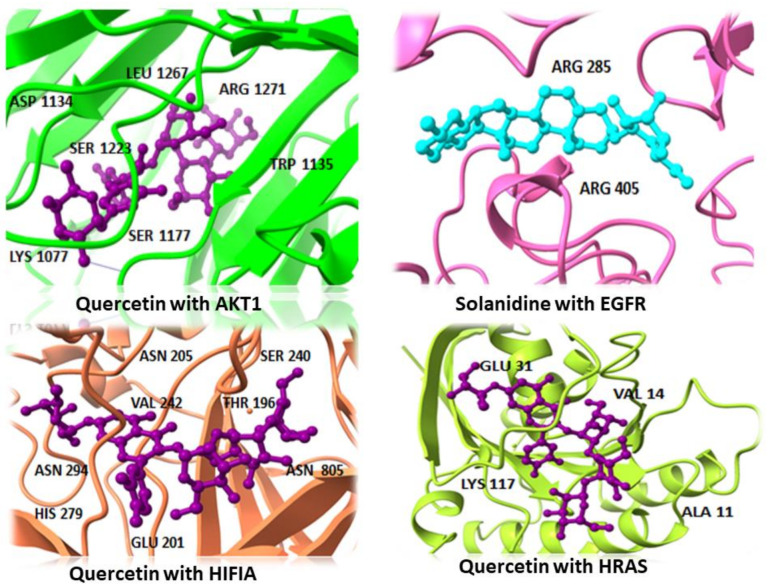
The docked complex of active constituents of *S. surattense*.

**Figure 7 molecules-27-06220-f007:**
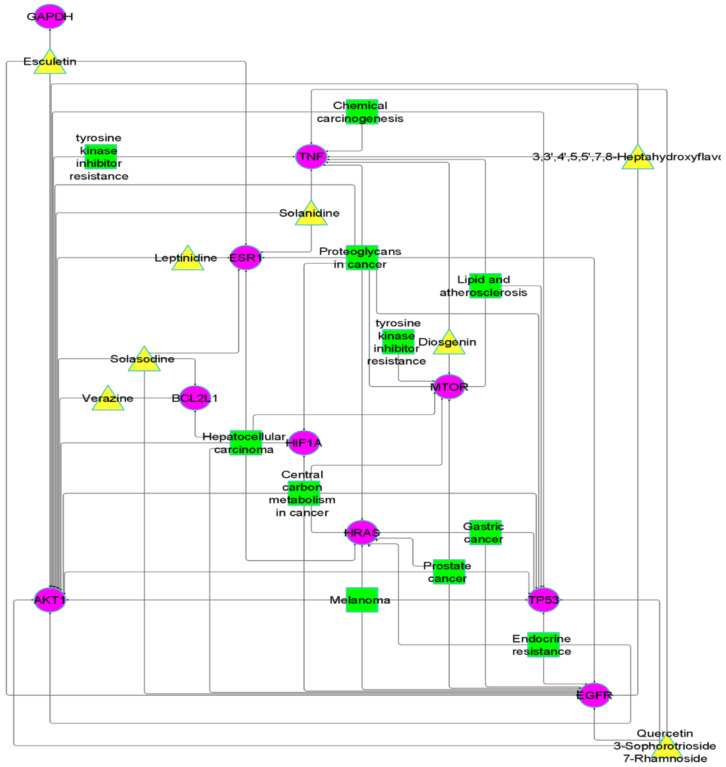
Pathways influenced by targets of *S. surattense*.

**Table 1 molecules-27-06220-t001:** Properties of active compounds.

Compound Name	MW	nHBD	nHBA	LogP	Lipinksi Rule
3,3′,4′,5,5′,7,8-Heptahydroxyflavone	390.1	3	9	2.6	Accepted
Leptinidine	413.33	2	3	4.6	Accepted
3′,4′,5,5′,7-Pentahydroxy-3-methoxy flavone	332.05	5	8	2.11	Accepted
Rishitin	222.16	2	2	2.38	Accepted
Spirosolan-3-ol	415.35	2	3	4.9	Accepted
Spirosol-5-en-3-ol	413.33	2	3	5.0	Accepted
Spirost-5-ene-3,25-diol	430.31	2	4	4.4	Accepted
Spirost-5-en-3-ol	414.31	1	3	5.5	Accepted
11-Spirovetivene-2,14-diol	236.18	1	2	2.2	Accepted
Verazine	413.33	2	3	5.1	Accepted
Campesterol	400.37	1	1	5.5	Accepted
Coumarin	146.04	0	1	1.6	Accepted
Diosgenin	414.31	1	3	5.5	Accepted
Esculetin	178.03	2	4	0.9	Accepted
Esculin	340.08	5	9	−0.6	Accepted
Methyl caffeate	194.06	2	4	1.9	Accepted
Solanidine	397.33	1	2	5.6	Accepted
Solanocapsine	430.36	4	4	5.0	Accepted
Solasodine	413.33	2	3	5.2	Accepted
Tomatidinol	413.33	2	3	5.3	Accepted
Solanine	867.5	9	16	2.0	Rejected
Quercetin 3-Galactoside 7-Rhamnoside	610.15	10	16	−0.8	Rejected
Quercetin 3-Sophorotrioside 7-Rhamnoside	934.26	16	26	−2.9	Rejected

**Table 2 molecules-27-06220-t002:** Binding energies and interaction of potential docked compounds.

Target Proteins	Compounds	Binding Affinity (kcal/mol)	RMSD	Interacting Residues
AKT1	Quercetin	−15.833	1.54	SER:1177, ASP:1134, TRP:1135, SER:1223, LYS:1077, LEU:1267, ARG:1271
	Solanidine	−14.353	1.62	SER:1177, VAL:1181, ASP:1229
	HeptahydroxyFlavone	−13.48	0.81	SER:1177, VAL:1220, THR:1265
BCL2L1	Esculetin	−7.054	1.81	GLN:26, GLN:160
	HeptahydroxyFlavone	−9.353	1.71	GLU:158, GLN:160
	Quercetin	−7.396	2.79	GLN:26, GLU:158, SER:25, SER:23
EGFR	Solanidine	−15.81	2.37	ARG B:285, ARG B:405
	Quercetin	−15.154	2.09	ARG B:285, SER B:342, ARG B:405
	HeptahydroxyFlavone	−13.245	1.38	SER B:11, ARG B:285, ARG B:405
ESR	Quercetin	−7.167	2.44	ASP 351
	HeptahydroxyFlavone	−12.73	1.27	ASP:351
	Leptinidine	−12.532	0.71	ASP:351
H1F1A	Quercetin	−22.917	2.68	ASN A:205, ASN A:294, THR A:196, SER A:240, VAL A:242, GLU A:201, HIS A:279ASN B:805
	Solanidine	−17.346	1.31	GLU A:201, GLN A:203, ASN A:205, TRP A:296, PRO A:274, TRP A:277
	HeptahydroxyFlavone	−17.46	0.81	LEU A:101, ASN B:803
HRAS	Verazine	−13.267	1.07	LYS:16, THR:35, GLN:61, LYS:117
	Quercetin	−20.618	1.91	ALA:11, VAL:14, GLU:31, LYS:117
	HeptahydroxyFlavone	−17.543	1.64	VAL:14, GLY:15, LYS:16, SER:17, LYS:117
MTOR	Quercetin	−12.382	1.81	LYS B:2046, TYR B:2089, GLY B:2093, ARG B:2043
	Solanidine	−8.277	1.15	ARG B:2043, ASN B:2044
	HeptahydroxyFlavone	−8.719	1.99	ARG B:2043,
TNF	Esculetin	−9.637	0.95	GLY B:122, GLY C:121
	HeptahydroxyFlavone	−5.584	1.94	TYR A:119, TYR B:119, GLY C:121

## Data Availability

Not applicable.

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
