# Peer review of "Integrated System Pharmacology Approaches to Elucidate Multi-Target Mechanism of Solanum surattense against Hepatocellular Carcinoma"

_molecules, 2022, doi:10.3390/molecules27196220_

Round 1

Reviewer 1 Report

Please correct the plant name in Solanum surattense.

The abbreviations have to be reported in extenso at their first appearance in the text.

Regarding the figures 1A-1B, please substitute anti-cancer effect with citotoxicity effect.

Please improve the quality of each subfigure 3.

Paragraph 4.2: How were the extraction conditions selected?

Paragraph 4.10: Please include the PDB code of the docked proteins.

Please format the references according to journal guidelines.

Author Response

  1. Please correct the plant name in Solanum surattense.

Thanks reviewer for valuable suggestion.The name of the herb Solanum surattense has been corrected throughout the manuscript.

  1. The abbreviations have to be reported in extenso at their first appearance in the text.

All the abbreviations are modified and given as full word extenso at their first appearance. All changes highlighted in the main manuscript  

  1. Regarding figures 1A-1B, please substitute anti-cancer effect with cytotoxicity effect.

Modified according to suggestion and more detail about these effects has been updated in the manuscript.

  1. Please improve the quality of each subfigure 3.

The resolution of subfigure 3 has been increased and a clear figure is now provided in the manuscript

  1. Paragraph 4.2: How were the extraction conditions selected?

The extraction conditions are elaborated in more detail in the same paragraph 4.2 (Extraction of plant material) & last sentence of first paragraph in discussion is added.

  1. Paragraph 4.10: Please include the PDB code of the docked proteins.

The PDB code of docked protein is already mentioned in the paragraph 2.7 but according to your suggestions now it is also included in paragraph 4.10 as  MTOR (1AUE), EGFR (1IVO), BCL2L1 (1R2E), ESR1 (1UOM), HRAS (4XVR), HIF1A (5JWP), TNF (5MU8)

  1. Please format the references according to journal guidelines.

References are now updated according to the journal format.

Reviewer 2 Report

Authors have reported the Integrated System pharmacology approaches to elucidate 2 multi-target mechanism of Solanum Surattense against Hepato- 3 cellular Carcinoma but some of the points need to be considered.

1. There are many reports for the use of herbal medicine against cancer  authors needs to go through it may be helpfull https://doi.org/10.3390/cancers14163898

2. What are the "research gaps" regarding the topic? It is better to provide a paragraph in introduction or above conclusion section to clearly describe what the research gaps

3. structure of the active should be given.

4.  Abstract is not properly written as the background of the study, methodology, result and conclusion need to be defined clearly

5.  If possible the docking may be validated using different software.

6.  The manuscript has many grammatical and syntax errors.

7. References are not as per journal guidelines

Author Response

  1. What are the “research gaps” regarding the topic?

Currently no comprehensive treatment method for hepatocellular carcinoma (HCC), therefore focused on systemic therapies, including drugs. Solanum surattense has been widely used in Asia to treat various diseases and cancer. However, there is no research on its core active component and targets. The main aim to find multi- targets mechanisms of Solanum surattense and its active ingredients against HCC

structure of the active should be given.

The structure of active compounds of S. surattense are now given in Figure.5

  1. Abstract is not properly written as the background of the study, methodology, result, and conclusion need to be defined clearly

The abstract part is modified now, and the proper background, methodology, result, and conclusion are elaborated in a modified version of the manuscript.

  1. If possible the docking may be validated using different software.

Yes, we perform the docking analysis with other tools such as MOE and Autodock. And the docked results like binding affinity and RMSD score are almost the same as we have done docking previously.

  1. The manuscript has many grammatical and syntax errors.

The complete manuscript has been revised and syntax and grammatical mistakes are now corrected in it.

  1. References are not as per journal guidelines

References are now updated according to the journal format.

Round 2

Reviewer 2 Report

it have been improved as per the suggestion

Round 3

Reviewer 2 Report

Suggestion  incorporated.

Thank you